# Prevalence and factors associated with hypertension among adults with and without HIV in Western Kenya

**Jerusha Nyabiage Mogaka**[1,2]*, **Monisha Sharma**[1], **Tecla Temu**[1], **Sarah Masyuko**[1,3],
**John Kinuthia**[2], **Alfred Osoti**[4,5], **Jerry Zifodya**[6], **Damalie Nakanjako**[7], **Anne Njoroge**[1],
**Amos Otedo**[3], **Stephanie Page**[1,8⊜], **Carey Farquhar**[1,8,9⊜]

1 Department of Global Health, University of Washington, Seattle, WA, United States of America,
2 Department of Research and Programs, Kenyatta National Hospital, Nairobi, Kenya, 3 Ministry of Health,
Nairobi, Kenya, 4 Department of Obstetrics and Gynecology, University of Nairobi, Nairobi, Kenya,
5 Department of Clinical Medicine and Therapeutics, College of Health Sciences, University of Nairobi,
Nairobi, Kenya, 6 Department of Medicine, Section of Pulmonary, Critical Care, & Environmental Medicine,
Tulane University, New Orleans, LA, United States of America, 7 Department of Medicine, School of
Medicine, Makerere University College of Health Sciences, Kampala, Uganda, 8 Department of Medicine,
University of Washington, Seattle, WA, United States of America, 9 Department of Epidemiology, University
of Washington, Seattle, WA, United States of America

⊜ These authors contributed equally to this work.
* jerunyags@gmail.com

## Abstract

### Introduction

The burden of cardiovascular disease (CVD) is increasing in sub-Saharan Africa with
untreated hypertension being a major contributing factor. Understanding the magnitude of
the problem and risk factors associated with HIV and long-term antiretroviral therapy (ART)
is critically important for designing effective programs for diagnosing and treating hyperten-
sion in Kenya.

### Methods

In this cross-sectional study, we enrolled 300 persons with HIV (PWH) on long term ART
(≥6 months) and 298 HIV-negative adults seeking care at the Kisumu County Hospital
between September 2017 and May 2018. Hypertension was defined as blood pressure of
≥140/90mmHg or a previous hypertension diagnosis. Multivariate regression was used to
assess the association between hypertension and HIV adjusting for age, sex, and known
CVD risk factors.

### Results

Overall prevalence of hypertension was 22%. PWH had a lower prevalence of hypertension
than HIV-negative persons (16% vs 27% respectively; p<0.002). In multivariate analyses,
persons with HIV were 37% less likely to have hypertension compared to HIV-negative indi-
viduals (adjusted prevalence ratio 0.63; 95% confidence interval: 0.46–0.86). Other factors
that were associated with hypertension in all participants included older age >40 years,

Science Center at San Antonio, UNITED STATES

**Data Availability Statement:** All relevant data are
within the manuscript.

**Funding:** This study was supported by US National
Institutes of Health (NIH), R21 TW010459 and the

Fogarty International Center (FIC) D43 TW009580. Monisha Sharma received support from NIMH K01MH115789. The funders had no role in study design, data collection and analysis, decision to publish, or preparation of the manuscript.

**Competing interests:** The authors declare that they have no conflicts of interest or competing financial interests.

body mass index (BMI) >25 kg/m$^2$ and low-density lipoproteins ≥130mg/dL. Among PWH, being older than 40 years and higher BMI >30 kg/m$^2$ were associated with hypertension.

## Conclusion

Prevalence of hypertension was high, affecting nearly one in every 4 adults, and associated with older age, higher BMI and high low-density lipoproteins. PWH on long-term ART had significantly lower prevalence of hypertension compared to HIV-negative individuals, potentially due to increased access to healthcare services and interaction with prevention messaging. Interventions to increase screening for and prevention of hypertension in the community for all adults are warranted.

## Introduction

Hypertension is a major modifiable risk factor for cardiovascular diseases (CVD) globally. In low- and middle-income settings, including sub-Saharan Africa (SSA), hypertension prevalence has been increasing rapidly over the past several decades. The World Health Organization (WHO) estimates that 46% of individuals >25 years in SSA have hypertension, with rising rates due to demographic transitions that have led to sedentary lifestyles, smoking, harmful alcohol use and consumption of processed foods [1–3]. Estimates of hypertension prevalence in Kenya are high (ranging from 12.6–36.9%) with higher rates in urban areas [1, 4, 5]. Older age, higher body mass index (BMI), alcohol consumption, cigarette smoking, and higher socioeconomic status have been associated with hypertension in previous studies in Kenya [5–7].

However, hypertension diagnosis and treatment are often delayed due to its asymptomatic nature, leading to increased risk of complications and mortality [8]. In SSA, screening, diagnosis, and treatment remain inadequate [9] and a recent study found that 40% of individuals with hypertension in East and West Africa were unaware of their status. The WHO 2017 report on non-communicable diseases (NCD) risk factors identified hypertension as the leading cause of death across income levels [10]. In 2015, hypertension caused an estimated 7.5 million deaths, accounting for 12.8% of all deaths globally [11]. In particular, sub-Saharan Africa (SSA) is facing a dual burden of communicable and non-communicable diseases, including CVD and cancers, with fewer resources for managing NCD [1, 12, 13].

The widespread use of antiretroviral therapy (ART) in SSA has resulted in a near normal life expectancy among persons with HIV (PWH); overall approximately 76% of PWH in SSA are virally suppressed [14]. This increased lifespan, however, may lead to an increased risk of NCD, including hypertension, due to the HIV virus and ART toxicity [14–17]. Studies on hypertension in PWH have shown varied results, some showing higher prevalence of hypertension while others showing no differences or lower prevalence of hypertension among PWH [18, 19]. The majority of studies have included PWH who are ART naïve or on ART but with poorly controlled viral loads compared to HIV-negative individuals in SSA [15, 18, 20, 21]. Data are lacking among PWH who are virally suppressed on ART. We sought to estimate the prevalence of hypertension among virally suppressed PWH on long-term ART compared to HIV-negative adults in western Kenya and identify factors associated with hypertension. These data can help guide prevention strategies and inform allocation of resources for integrated hypertension and HIV management.

## Materials and methods

### Study design and setting

We used data from a cross-sectional hospital-based study of 300 PWH and 298 HIV-negative adults, that assessed how CVD risk factors among PWH on ART compared to HIV negative individuals. Participants were enrolled between September 2017 and May 2018 at the Kisumu County Hospital (KCH) in Western Kenya. We selected KCH as our recruitment site since its patient population of 12,903 PWH covers a large catchment area around 59,000 which increases the likelihood of enrolling a representative sample. Although we recruited from a hospital, we enrolled HIV-negative participants from voluntary HIV counseling and testing (VCT) services as these individuals were likely representative of the healthy general population. We utilized stratified random sampling to ensure an equal number of male and females since male sex is associated with higher CVD risk.

Detailed descriptions of the recruitment strategy and study procedures are presented in the parent study [22]. Briefly, participants ≥30 years old, living within a 50 km radius of the hospital, and seeking routine services at KCH were eligible to participate in the study. Eligible PWH were in care at an HIV comprehensive care clinic and on ART for ≥6 months. HIV-negative persons were recruited from voluntary HIV testing and counseling services at KCH.

### Study procedures

We utilized a stratified random sampling technique to recruit eligible participants. Inclusion criteria for all participants were being ≥30 years old, living within a 50 km radius of the hospital, seeking routine services at KCH, and willing to provide informed consent. Eligibility criteria for PWH were being enrolled in care at an HIV comprehensive care clinic and on ART for ≥6 months. Eligibility criteria for HIV-negative persons were seeking voluntary HIV testing and counseling services at KCH and screening negative for a rapid HIV test. We excluded from the study individuals not willing to give informed consent or participate in any of the study procedures, not willing to screen for HIV for those who verbalized they were HIV negative and PWH who were on ART for <6 months.

We obtained ethical approval from the University of Washington Institutional Review Board and the University of Nairobi and Kenyatta National Hospital Ethics Review Committee. Written informed consent was obtained from all participants.

Trained nurse counselors collected data on sociodemographic and behavioral characteristics, HIV status and hypertension risk factors using a structured questionnaire adapted from the validated WHO STEPwise approach for chronic disease risk factors surveillance, modified to the Kenyan context [22, 23]. Blood pressure readings, anthropometric measurements and blood draws were performed the same day for participants who had fasted for at least 8 hours. Those who had not fasted were asked to return the following day for the venipuncture. Data abstraction from medical records was conducted for HIV-related variables such as time since HIV diagnosis, ART regimen type, Time since ART initiation, CD4 cell count and viral load suppression.

Fasting blood samples were collected in red top tubes, processed and serum was extracted stored at Kenya Medical Research Institute Centers for Disease Control and Prevention (KEMRI/CDC) lab in Kisumu, Kenya. Testing for CD4 T-cell count and viral load for PWH were collected in EDTA anticoagulant tubes, centrifuged to extract plasma and processed locally at the KEMRI/CDC lab. A detailed description of laboratory procedures has been previously described [24].

## Definition of variables

Hypertension was classified according to the Kenyan guidelines [25]. which are adopted from the Eighth Joint National Committee on Prevention, Detection, Evaluation, and Treatment of high blood pressure (JNC VIII) [26].

The primary outcome, hypertension, was defined as mean systolic blood pressure ≥140 mm Hg or diastolic blood pressure ≥90mm Hg, or self-report of previous hypertension diagnosis by a health care provider and/or currently taking anti-hypertensive drugs within the last two weeks. Two blood pressure readings and pulse rate were taken for each arm, 5 minutes apart using a digital sphygmomanometer (CH 453, Omron Health). The first set of readings were dropped with an average of the remaining considered in the final analysis. Blood pressure control was defined as being on treatment within the last two weeks and having a mean blood pressure reading of less than 140/90mm Hg.

Body Mass Index (BMI) was computed from weight (kilograms) and height (meters). BMI was classified as underweight ($<18$kg/m$^2$), normal weight ($18–25$ kg/m$^2$), overweight ($25–30$ kg/m$^2$) or obese ($>30$ kg/m$^2$). Adequate physical activity was defined as engaging in at least 150 minutes of moderate work or sports or at least 75 minutes of vigorous intensity work or sport per week [27]. Current alcohol intake and cigarette smoking were defined as use within the past 30 days. At least 5 servings of fruit and vegetables per day were considered adequate. Abdominal obesity was defined as a waist circumference of $>88$ cm in females and $>94$ cm in males, while central obesity was defined as a hip-waist ratio of $>0.80$ in females and $>0.90$ in males based on the WHO recommendations [28].

Cut-offs for total cholesterol and triglycerides levels were 200 mg/dL and 150 mg/dL respectively. Elevated low-density lipoprotein (LDL) was defined as LDL of $>130$ mg/dL while high-density lipoprotein (HDL) cutoffs were $<40$ mg/dL in males and $<50$ mg/dL in females. Elevated fasting blood sugar (FBS) was defined as a blood glucose level $>100$ mg/dL. A viral load of $<1000$ copies/mL was defined as viral suppression, while viral load $<50$ copies/mL was defined as undetectable based on the current Kenyan guidelines [29].

## Statistical analyses

Continuous variables were analyzed using student t-test and categorical variables were evaluated using chi-square tests. Univariate and multivariable models were fit using poisson regression model with robust standard errors to assess the association between HIV status, and traditional risk factors (diabetes, dyslipidemia, smoking, elevated BMI) with hypertension. We conducted sub-analysis limited to PWH, including HIV specific characteristics such as viral load, CD4 count, ART regimen and years on ART. We computed 95% CI with p-values $<0.05$ considered statistically significant. All analyses were conducted using Stata version 15.0 (Stata Corp. College Station TX).

## Results

### Participant characteristics

Of the 600 participants enrolled in the parent study, 598 participants were included in the present analysis: 300 PWH and 298 HIV-negative participants, 50% were female. Two participants were excluded due to missing data on HIV status and laboratory results. The median age was 45 (interquartile range [IQR] 39.5, 53) years in PWH and 40 (IQR: 31, 55) for the HIV-negative participants (**Table 1**). Majority (86%) of the participants had completed at least primary level education, and two-thirds (69%) were formally employed. Mean BMI was 23.3 (95% [confidence interval] CI: 22.7, 24.7) in PWH and 25.1 (95% CI: 24.4, 25.8) in HIV-negative

**Table 1. Participant characteristics stratified by HIV status[§].**

| | Overall | | HIV +, | | HIV -, | | p-value |
|---|---|---|---|---|---|---|---|
| | N = 598 | | N = 300 | | N = 298 | | |
| | n (%) | | n (%) | | n (%) | | |
| Sex Male | 299 | (50) | 150 | (50) | 149 | (50) | 1.00 |
| **Age groups (years)[δ]** | | | | | | | |
| 30–39 | 213 | (36) | 75 | (25) | 138 | (46) | |
| 40–50 | 303 | (50) | 195 | (65) | 108 | (36) | <0.001 |
| >50 | 82 | (14) | 30 | (10) | 52 | (17) | |
| **Level of Education** | | | | | | | |
| None | 85 | (14) | 39 | (13) | 46 | (15) | |
| Primary level | 227 | (38) | 130 | (43) | 97 | (33) | 0.025 |
| More than primary level | 286 | (48) | 131 | (44) | 155 | (52) | |
| **BMI (kg/m$^2$)[δ]** | | | | | | | |
| <18(underweight) | 44 | (7) | 27 | (9) | 17 | (6) | |
| ≥18-<25 (normal) | 349 | (58) | 192 | (64) | 157 | (52) | |
| ≥25-<30 (overweight) | 128 | (21) | 57 | (19) | 71 | (24) | 0.001 |
| ≥30 (obese) | 77 | (13) | 24 | (8) | 53 | (18) | |
| **Marital Status** | | | | | | | |
| Currently married | 440 | (74) | 205 | (68) | 235 | (79) | |
| Never married | 38 | (6) | 15 | (5) | 23 | (8) | <0.001 |
| Divorced/Separated/Widowed | 120 | (20) | 80 | (27) | 40 | (3) | |
| **Occupation** | | | | | | | |
| Employed | 411 | (69) | 208 | (69) | 203 | (68) | |
| Casual Laborer | 100 | (17) | 52 | (17) | 48 | (16) | 0.68 |
| Unemployed | 87 | (15) | 40 | (13) | 47 | (16) | |
| **Alcohol consumption[δ]** | | | | | | | |
| Never | 401 | (67) | 204 | (68) | 197 | (66) | |
| Previous | 123 | (21) | 64 | (21) | 59 | (20) | 0.43 |
| Current | 74 | (12) | 32 | (11) | 42 | (14) | |
| **Smoker** | | | | | | | |
| Never | 524 | (88) | 261 | (87) | 263 | (88) | |
| Previous | 45 | (7) | 27 | (9) | 18 | (6) | 0.26 |
| Current | 29 | (5) | 12 | (4) | 17 | (6) | |
| **Insufficient fruit and vegetable servings** | 551 | (93) | 282 | (94) | 269 | (91) | 0.10 |
| **Insufficient physical Activity** | 258 | (43) | 113 | (38) | 144 | (48) | 0.01 |
| **Central obesity** | 292 | (49) | 156 | (52) | 136 | (46) | 0.12 |
| **Abdominal obesity** | 140 | (23) | 56 | (19) | 84 | (28) | 0.006 |
| **Lipid profile[Ψ]** | | | | | | | |
| High total cholesterol (≥ 200 mg/dL) | 97 | (17) | 52 | (18) | 45 | (16) | 0.6 |
| High triglycerides (≥150 mg/dL) | 48 | (9) | 30 | (10) | 18 | (7) | 0.09 |
| High LDL (≥130 mg/dL) | 75 | (13) | 36 | (13) | 39 | (14) | 0.59 |
| Low HDL | 167 | (28) | 75 | (25) | 92 | (31) | 0.11 |
| **High FBS (>100 mg/dL) [Ψ]** | 24 | (4) | 7 | (2) | 17 | (6) | 0.031 |
| **HIV related characteristics** | | | | | | | |
| Viral load - Undetectable | | | 213 | (71) | | | |
| - Suppressed | | | 285 | (96) | | | |
| ART regimen - PI based | | | 40 | (13) | | | |

(*Continued*)

**Table 1.** (Continued)

| | Overall | | HIV +, | | HIV -, | | p-value |
|---|---|---|---|---|---|---|---|
| | N = 598 | | N = 300 | | N = 298 | | |
| | n (%) | | n (%) | | n (%) | | |
| - Non-PI based | | | 260 | (87) | | | |

[§]HIV+: Persons With HIV, HIV-: HIV-negative individuals. BMI-body mass index. Inadequate fruit and vegetable servings: Less than 5 servings per day. Physical inactivity: < 150 minutes of moderate intensity physical activity or < 75minutes of vigorous physical activity per week. Central obesity: waist-hip ratio >0.90 males and >0.80 females. Abdominal obesity: waist circumference >94cm in men and 88cm in female. LDL—low density lipoproteins, HDL–high density lipoprotein (low HDL: <40 mg/dL in male and <50 mg/dL in female. FBS-fasting blood sugar. Undetectable viral load-viral load <50copies/mL. Virally suppressed: viral load <1000copies/mL. PI-Protease based.

[δ]Due to rounding, some proportions do not add up to 100%.

[Ψ]N = 564

participants (p<0.001). Abdominal obesity was less prevalent in PWH compared to HIV-negative participants (19% vs 28% respectively, p = 0.01) while PWH were more likely to be physically active than HIV-negative participants (p = 0.01). Overall, current alcohol consumption (12%) and smoking prevalence (5%) were low with no significant differences by HIV status.

Among PWH, the median (IQR) time since diagnosis of HIV and ART duration was 9 (5,11) years and 8 (4,10) years, respectively. The median (IQR) current and nadir CD4 cell count was 512 (364, 666) cells/mm$^3$ and 369 (215, 563) cells/mm$^3$ respectively, while 96% were virally suppressed. Most PWH (87%) were on first line ART (non-protease inhibitor-based regimen).

## Hypertension prevalence, awareness, treatment, and control

The overall prevalence of hypertension was 22% (n = 129), with 16% of PWH and 27% of HIV-negative participants being hypertensive (p<0.002). Of the 129 individuals with hypertension, 71 (55%) reported a previous diagnosis of hypertension. A new diagnosis of hypertension was reported in 43% of PWH compared to 57% of the HIV-negative participants (p = 0.26). Only one-third (24%) reported taking antihypertensive medications in the past two weeks. Seven (27%) of those on medication met the criteria for blood pressure control.

## Risk factors of hypertension

Compared to normotensive participants, individuals with hypertension were more likely to be >40 years of age (84% vs 59%; p<0.001), have a BMI >25 kg/m$^2$ (58% vs 28; p<0.001) (Fig 1) and to be HIV-negative (62% vs 46%; p = 0.002) with no significant sex differences (Table 2). There were no significant differences in adequate nutrition intake, harmful smoking and alcohol consumption by hypertension status. Hypertension prevalence was higher in participants with abdominal obesity (45% vs. 17%, p<0.001) and central obesity (61% vs. 45%, p = 0.001) compared to non-obese participants. Individuals with hypertension were more likely to have elevated total cholesterol, high low-density lipoproteins (LDL), and high fasting blood sugar (FBS) compared to the normotensive participants (Table 2).

Next, we evaluated covariates associated with hypertension in univariate and multivariable analyses (Table 3). In univariate analyses, HIV status (Prevalence Ratio [PR] 0.61, 95% CI 0.44, 0.84), age >40 years, BMI >25 kg/m$^2$/, education level, elevated LDL (PR 2.58, 95% CI 1.88, 3.53), and elevated FBS (PR 2.07, 95% CI 1.25, 3.42) were associated with hypertension. In multivariable analyses, PWH were 37% less likely to have hypertension compared to HIV-negative individuals (aPR 0.63, 95% CI: 0.46, 0.86). Additionally, risk of hypertension

## A. By Age Categories

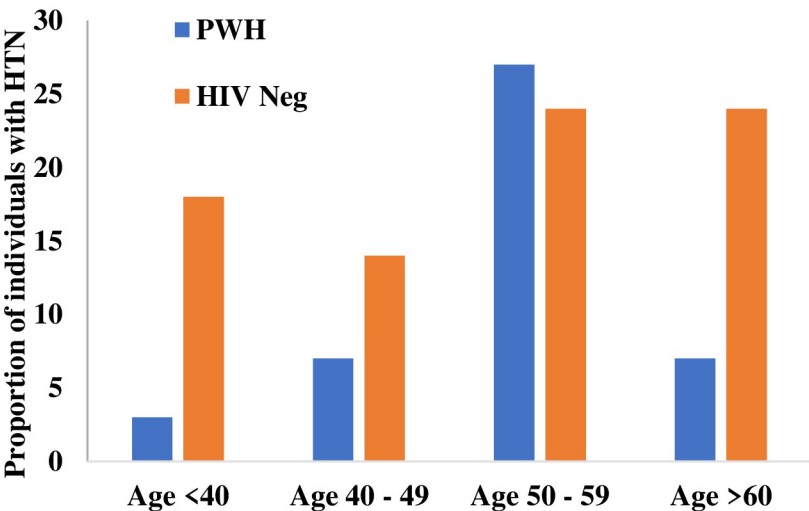

## B. By BMI Categories

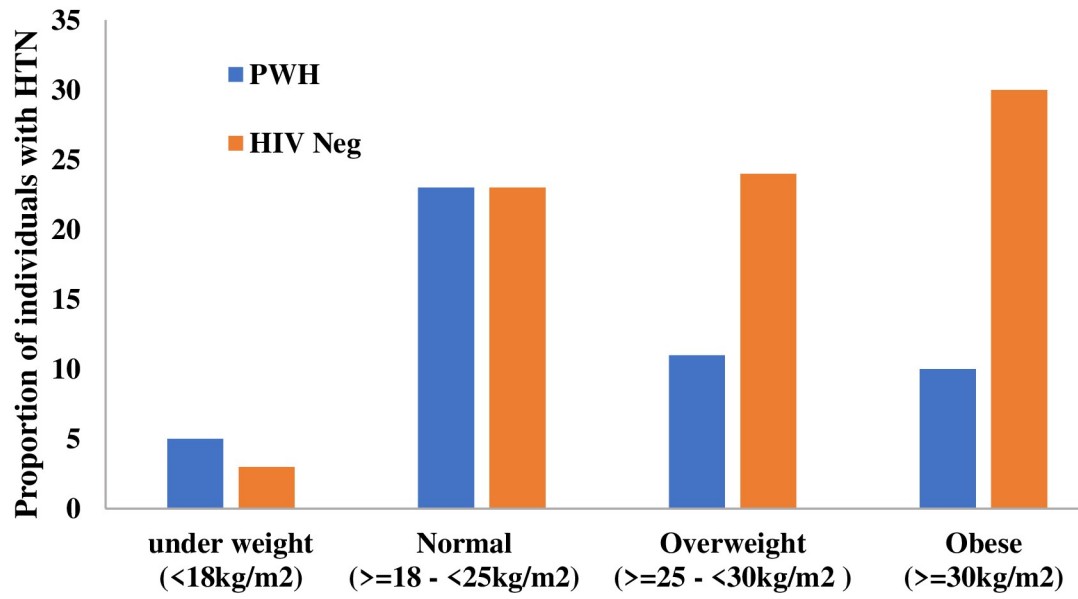

**Fig 1. Prevalence of hypertension among PWH and HIV negative adults.**

increased with age in a dose-response manner; the risk was 2.74 higher among persons who were age 40–50 years (95% CI: 1.71, 4.38) and 3.87 higher among those >50 years (95% CI: 2.19, 6.82) compared to those age 30–40 years. Similarly, the risk of hypertension was greater in those with higher BMI; individuals who were overweight (25–30 kg/m$^2$) were twice as likely to have hypertension (aPR 1.87, 95% CI: 1.26, 2.79) while obese participants (>30 kg/m$^2$) were

**Table 2. Characteristics of participants stratified by hypertension status[†].**

| | Hypertension | | No hypertension | | p-value |
|---|---|---|---|---|---|
| | n (%) | | n (%) | | |
| **Age groups (years)[δ]** | | | | | |
| 30–39 | 21 | (16) | 192 | (41) | |
| 40–50 | 77 | (60) | 226 | (48) | < 0.001 |
| >50 | 31 | (24) | 51 | (11) | |
| **Gender** | | | | | |
| Female | 72 | (56) | 227 | (48) | 0.14 |
| Male | 57 | (44) | 242 | (52) | |
| **Level of Education** | | | | | |
| None | 26 | (20) | 59 | (13) | |
| Primary level | 44 | (34) | 183 | (39) | 0.09 |
| Secondary level and above | 59 | (46) | 227 | (48) | |
| **BMI (kg/m$^2$)[δ]** | | | | | |
| <18(underweight) | 8 | (6) | 36 | (8) | |
| ≥18 - <25 (normal) | 46 | (36) | 303 | (65) | < 0.001 |
| ≥25 - <30 (overweight) | 35 | (27) | 93 | (20) | |
| ≥30 (obese) | 40 | (31) | 37 | (8) | |
| **HIV Status** | | | | | |
| PWH | 49 | (38) | 251 | (54) | 0.002 |
| HIV negative | 80 | (62) | 218 | (46) | |
| **Inadequate fruits and vegetable servings** | 118 | (93) | 433 | (93) | 0.89 |
| **Alcohol consumption** | | | | | |
| Never | 94 | (73) | 307 | (65) | 0.17 |
| Previous | 19 | (15) | 104 | (22) | |
| Current | 16 | (12) | 58 | (13) | |
| **Smoker** | | | | | |
| Never | 116 | (90) | 408 | (87) | |
| Previous | 5 | (4) | 40 | (9) | 0.16 |
| Current | 8 | (6) | 21 | (4) | |
| **Insufficient Physical activity** | 78 | (60) | 250 | (53) | 0.15 |
| **Central obesity** | 79 | (61) | 213 | (45) | 0.001 |
| **Abdominal obesity** | 58 | (45) | 82 | (17) | < 0.001 |
| **Lipid profile[Ψ]** | | | | | |
| High total cholesterol (≥ 200 mg/dL) | 38 | (32) | 59 | (13) | < 0.001 |
| High triglycerides (≥150 mg/dL) | 15 | (13) | 33 | (7) | 0.08 |
| High LDL (≥130 mg/dL) | 34 | (28) | 41 | (9) | < 0.001 |
| Low HDL | 39 | (30) | 128 | (27) | 0.51 |
| **High FBS (>100 mg/dL)[Ψ]** | 10 | (8) | 14 | (3) | 0.01 |

[†] BMI-body mass index. Physical inactivity: < 150 minutes of moderate intensity physical activity or < 75minutes of vigorous physical activity per week. Central obesity: waist-hip ratio > 0.90 males and > 0.80 females. Abdominal obesity: waist circumference > 94cm in men and >88cm in female. LDL—low density lipoproteins, HDL–high density lipoprotein (low HDL: <40 mg/dL in male and <50 mg/dL in female. FBS-fasting blood sugar.
[δ]Due to rounding some proportion do not add up to 100%.
[Ψ]N = 564.

2.83 times more likely to have hypertension (95% CI: 1.91, 4.17) compared to those with normal weight (18–25 kg/m$^2$).

**Table 3. Univariate and multivariate associations between risk factors and hypertension∞.**

| | | Univariate PR | 95% CI | p-value | Multivariate aPR | 95% CI | p-value |
|---|---|---|---|---|---|---|---|
| **HIV Status:** | Negative | 1.00 | | | 1.00 | | |
| | Positive | 0.61 | (0.44–0.84) | 0.002 | 0.63 | (0.46–0.86) | 0.004 |
| **Gender**: | Female | 1.00 | | | | | |
| | Male | 0.79 | (0.58–1.08) | (0.14) | | | |
| **Age: (years)** | <40 | 1.00 | | | 1.00 | | |
| | 40–50 | 2.58 | (1.64–4.04) | < 0.001 | 2.74 | (1.71–4.38) | < 0.001 |
| | >50 | 3.83 | (2.34–6.27) | < 0.001 | 3.87 | (2.19–6.82) | < 0.001 |
| **BMI (kg/m$^2$):** | 18–24 | 1.00 | | | 1.00 | | |
| | <18 | 1.38 | (0.70–2.7) | 0.36 | 1.42 | (0.74–2.76) | 0.29 |
| | 25–30 | 2.07 | (1.40–3.07) | < 0.001 | 1.87 | (1.26–2.79) | 0.002 |
| | >30 | 3.94 | (2.79–5.56) | < 0.001 | 2.83 | (1.91–4.17) | < 0.001 |
| **Level of Education** | | | | | | | |
| | None | 1.00 | | | 1.00 | | |
| | Primary level | 0.63 | (0.44–0.92) | 0.03 | 0.89 | (0.57–1.37) | 0.58 |
| | ≥ Secondary level | 0.67 | (0.46–0.92) | 0.05 | 0.87 | (0.58–1.30) | 0.50 |
| **High triglycerides levels** | | 1.54 | (0.98–2.42) | 0.06 | 1.03 | (064–1.68) | 0.88 |
| **High low-density lipoprotein** | | 2.58 | (1.88–3.53) | < 0.001 | 1.67 | (1.18–2.33) | 0.003 |
| **High fasting blood sugar** | | 2.07 | (1.25–3.42) | 0.005 | 1.36 | (0.84–2.22) | 0.22 |
| **Alcohol intake** | | | | | | | |
| | Never | 1.00 | | | | | |
| | Previous | 0.65 | (0.62–1.37) | 0.07 | 0.66 | (0.42–1.01) | 0.06 |
| | Current | 0.92 | (0.43–1.02) | 0.74 | 1.16 | (0.69–1.94) | 0.57 |
| **Smoker** | | | | | | | |
| | Never | 1.00 | | | | | |
| | Previous | 0.50 | (0.22–1.17) | 0.11 | | | |
| | Current | 1.25 | (0.68–2.30) | 0.48 | | | |
| **Insufficient physical activity** | | 1.26 | (0.92–1.72) | 0.15 | | | |
| **Insufficient fruit &vegetable servings** | | 1.05 | (0.57–1.92) | 0.88 | | | |

∞Adjusting for age, BMI, education, high triglycerides, high low-density lipoproteins, high fasting blood sugar and alcohol intake, the likelihood of having hypertension in PWH is 37% lower than the HIV-negative individuals. aPR: adjusted prevalence ratio. High triglycerides levels: ≥150mg/dL. High low-density lipoprotein: ≥130mg/dL. High fasting blood sugar: >100mg/dL

Restricting our analysis to PWH, age >40 years and BMI >30 kg/m$^2$ (aPR 3.39, 95% CI: 1.89, 6.10) were associated with hypertension in multivariable analyses. Hypertension was not associated with HIV specific characteristics (**Table 4**).

## Discussion

In this study of Kenyan adults, PWH had a lower hypertension prevalence than the HIV-negative individuals. PWH were less likely to be overweight or obese and more likely to be involved in recommended physical activity as compared to the HIV-negative participants, potentially explaining this difference. A quarter of the participants had hypertension with more than half of these being a new diagnosis. Hypertensive PWH were more likely to have a been previously diagnosed with hypertension than their HIV-negative counterparts. Advancing age, increasing BMI, and elevated LDL were associated with hypertension.

The higher prevalence of hypertension among the HIV-negative individuals compared to PWH, did not support the hypothesis that PWH are at increased risk of hypertension because

**Table 4. Univariate and multivariate associations between risk factors and hypertension among PWH[∞].**

| | | Univariate PR | 95% CI | p-value | Multivariate aPR | 95% CI | p-value |
|---|---|---|---|---|---|---|---|
| **Age:** | <40 years | 1.00 | | | 1.00 | | |
| | 40–50 years | 5.00 | (1.59–15.72) | 0.006 | 4.75 | (1.50–15.07) | 0.008 |
| | >51 years | 5.83 | (1.61–21.12) | 0.007 | 6.32 | (1.69–23.54) | 0.006 |
| **BMI** | 18–25 | 1.00 | | | 1.00 | | |
| **(kg/m²):** | <18 | 1.55 | (0.64–3.73) | 0.33 | 1.37 | (0.57–3.26) | 0.48 |
| | 25–30 | 1.61 | (0.83–3.10) | 0.15 | 1.65 | (0.88–3.11) | 0.12 |
| | >30 | 3.48 | (1.89–6.40) | < 0.001 | 3.39 | (1.89–6.10) | <0.001 |
| **Level of Education** | | | | | | | |
| | None | 1.00 | | | | | |
| | Primary level | 1.44 | (0.59–3.52) | 0.43 | | | |
| | ≥Secondary level | 1.19 | (0.48–2.97) | 0.71 | | | |
| **Alcohol intake** | | | | | | | |
| | Never | 1.00 | | | | | |
| | Previous | 0.55 | (0.24–1.24) | 0.15 | | | |
| | Current | 1.46 | (0.74–2.86) | 0.27 | | | |
| **Smoker** | | | | | | | |
| | Never | 1.00 | | | | | |
| | Previous | 0.21 | (0.31–1.50) | 0.12 | | | |
| | Current | 1.45 | (0.52–4.01) | 0.47 | | | |
| **Insufficient physical activity** | | 1.02 | (0.58–1.78) | 0.95 | | | |
| **Insufficient fruit & vegetable servings** | | 0.68 | (0.28–1.67) | 0.40 | | | |
| **Detectable Viral load** | | | | | | | |
| | ≥50copies/mL³ | 1.00 | | | | | |
| | <50copies/mL³ | 1.53 | (0.72–3.26) | 0.27 | | | |
| **Current CD4 count** | | | | | | | |
| | ≥500 cells/mm³ | 1.00 | | | | | |
| | <500 cells/mm³ | 1.00 | (0.60–1.69) | 0.97 | | | |
| **Nadir CD4 count** | | | | | | | |
| | ≥500 cells/mm³ | 1.00 | | | | | |
| | <500 cells/mm³ | 0.93 | (0.55–1.58) | 0.78 | | | |
| **ART regimen** | | | | | | | |
| | Non-PI based | 1.00 | | | | | |
| | PI based | 0.74 | (0.31–1.75) | 0.49 | | | |
| **ART duration (years)** | | 1.03 | (0.97–1.09) | 0.34 | | | |

[∞]PWH: persons with HIV. aPR: adjusted prevalence ratio. PI: protease inhibitors. Undetectable viral load-viral load < 50copies/mL. Physical inactivity: <150 minutes of moderate intensity physical activity or < 75minutes of vigorous physical activity per week. Inadequate fruit and vegetable servings: Less than 5 servings per day

of increased the HIV virus and ART use leading to immune activation [30, 31]. Previous studies assessing hypertension by HIV status have yielded conflicting results [20, 21, 32] Population-based surveys conducted in Uganda and North Tanzania also found higher rates of hypertension in HIV-negative individuals compared to PWH (14% vs 11% in Uganda and 8.2% vs 5.3% in Tanzania) [20, 21]. The authors of these studies suggest that HIV-negative individuals may have more anxiety related to seeing a healthcare provider which may lead to a higher blood pressure measurement compared to PWH who have greater contact with the healthcare system. They also posit that hypertension prevalence in PWH is attributed to survival bias if persons with both HIV and hypertension may have higher death rates [21].

Anxiety is unlikely to influence our study findings as blood pressures were checked after 5 minutes of rest and we discarded the first measured blood pressure. It is possible that early initiation of ART after HIV diagnosis combined with high adherence can reduce the risk of hypertension among PWH. There is also a possibility that PWH have better health habits compared to HIV-negative individuals due to regular interaction with health care workers, where PWH are more likely to be monitored for risk factors related to hypertension. Traditional risk factors did play a significant role in the higher prevalence of hypertension. HIV-negative individuals were more likely to be obese and overweight and less likely to meet the recommended physical activity.

The hypertension prevalence found in our study was slightly lower than that reported in the Kenyan national survey on non-communicable diseases (24.5%) [23] but comparable to other studies conducted in Kenya [5, 33, 34] and in SSA [20]. Rates of undiagnosed hypertension were high in both groups; however, they were significantly higher in HIV-negative adults. A possible explanation is that PWH interact more with the healthcare workers due to the scheduled ART refill visits therefore more likely to be screened for hypertension. We also found a low number of individuals previously diagnosed with hypertension being on treatment or achieving blood pressure control in PWH and HIV-negative participants. It is imperative to leverage on evidence-based interventions that have previously worked to increase opportunities for diagnosis of hypertension and both uptake and continuation of treatment in hypertensive patients. These interventions include integration of hypertension related health education in community-based service provision and other hospital-based service delivery points. Similar proven interventions have been successful in increasing population coverage of HIV testing, treatment, and adherence.

Less than half of the individuals with previously diagnosed hypertension were taking anti-hypertensive medication, and among those on treatment, many had not achieved hypertension control. Surprisingly, most of the individuals on treatment for hypertension were HIV-negative. This highlights a gap in follow-up of PWH with hypertension as a comorbidity, despite frequent healthcare visits. Similarly, the national survey in Kenya found that approximately 40% of those with known hypertension were on treatment, 49% of whom had achieved blood pressure control [8, 34]. These studies highlight the gap between hypertension screening and linkage to treatment. Interventions are urgently needed to increase treatment uptake and monitor treatment adherence to reduce the risk of complications related to uncontrolled hypertension.

Hypertension was associated with older age and higher BMI. Surprisingly, we did not find an association between other known risk factors of hypertension (physical activity, diet, socio-economic status, alcohol intake and cigarette smoking) and hypertension. This may be due to imprecise measurements of these factors possibly due to social desirability bias due to self-report. These results are partially consistent with the 2015 national survey on prevalence and determinants of hypertension in Kenya, highlighting the potential of this phenomenon affecting reporting [23].

Results of our sub-analysis restricted to PWH found that ART regimen, duration of time on ART, viral load, current and nadir CD4 T cell count viral load were not associated with hypertension, which parallels findings from previous studies [20, 35]. This may be because majority of PWH in our sample had well controlled HIV with high CD4 count and were virally suppressed. Similar to other studies, we found hypertension to be associated with elevated LDL [35].

Our study has several strengths. We enrolled a large sample of both PWH and HIV-negative well-matched controls from the same region. PWH in our study population were stable on ART with most achieving viral suppression showing effective management. Few studies in the

region have compared these two populations after the roll out of test and treat programs in Kenya which has likely increased the linkage to care of PWH. Thus our population of PWH is likely representative of current in the greater Kenyan population. Our study also has limitations: this is a cross-sectional study; thus, we were unable to assess temporal relationships. Additionally, there may be selection bias since the HIV-negative individuals from the community who came to the hospital for HIV screening may differ from those in the community. However, we recruited HIV-negative persons seeking routine HIV testing services at the voluntary HIV testing site, which is likely representative of healthy individuals in the community. PWH in our study were on long-term ART and nearly all were virally suppressed, therefore, we cannot assess the association between HIV and hypertension among PWH who are ART naïve or not suppressed on ART. Further, due to incomplete data on socioeconomic status (which has been associated with hypertension in literature), residual confounding is possible. Finally, diagnosis of hypertension requires multiple measurements however due to the nature of the study, we only had measurements for one day. This may overestimate the true prevalence of hypertension in this cohort.

## Conclusion

We found a high prevalence of hypertension and a large proportion of undiagnosed hypertension in both PWH and HIV-negative adults. In this study, PWH on long-term ART with viral suppression had a lower prevalence of hypertension compared to HIV-negative individuals. Strengthening hypertension screening programs in both PWH and HIV-negative adults, establishing strong referral systems and linkage to care is crucial to averting adverse outcomes related to undiagnosed hypertension.

## Acknowledgments

We thank the study team, study participants, collaboration from University of Washington, USA, University of Makerere, Kampala, Uganda, Ministry of Health, Kenya, Kenyatta National Hospital, Kenya and Kisumu County Hospital, Kenya.

## Author Contributions

**Conceptualization:** Tecla Temu, Sarah Masyuko, Alfred Osoti, Anne Njoroge, Stephanie Page, Carey Farquhar.

**Data curation:** Jerusha Nyabiage Mogaka, Sarah Masyuko.

**Formal analysis:** Jerusha Nyabiage Mogaka.

**Funding acquisition:** Carey Farquhar.

**Methodology:** Tecla Temu, Sarah Masyuko, Anne Njoroge, Stephanie Page, Carey Farquhar.

**Project administration:** Jerusha Nyabiage Mogaka.

**Supervision:** Monisha Sharma, Tecla Temu, Sarah Masyuko, Stephanie Page, Carey Farquhar.

**Visualization:** Jerusha Nyabiage Mogaka.

**Writing – original draft:** Jerusha Nyabiage Mogaka.

**Writing – review & editing:** Monisha Sharma, Tecla Temu, John Kinuthia, Alfred Osoti, Jerry Zifodya, Damalie Nakanjako, Anne Njoroge, Amos Otedo, Stephanie Page, Carey Farquhar.

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
