## [Decision Letter · Decision Letter 0]

16 Jun 2021

PONE-D-21-10329

Hypertension prevalence and correlates among adults with and without HIV in Western Kenya

PLOS ONE

Dear Dr. Jerusha Nyabiage Mogaka,

Thank you for submitting your manuscript to PLOS ONE. After careful consideration, we feel that it has merit but does not fully meet PLOS ONE’s publication criteria as it currently stands. Therefore, we invite you to submit a revised version of the manuscript that addresses the all the points raised by both reviewers, which I agreed. 

We look forward to receiving your revised manuscript.

Kind regards,

Weijing He, M.D.

Academic Editor

PLOS ONE

Journal Requirements:

2. In your Methods section, please provide additional information about the participant recruitment method and the demographic details of your participants. Please ensure you have provided sufficient details to replicate the analyses such as: a) the recruitment date range (month and year), b) a description of any inclusion/exclusion criteria that were applied to participant recruitment, c) a table of relevant demographic details, d) a statement as to whether your sample can be considered representative of a larger population, e) a description of how participants were recruited, and f) descriptions of where participants were recruited and where the research took place. Please note that according to our policies, if materials, methods, and protocols are well established, authors may cite articles where those protocols are described in detail, but the submission should include sufficient information to be understood independent of these references (https://journals.plos.org/plosone/s/submission-guidelines#loc-materials-and-methods).

3. Please amend the manuscript submission data (via Edit Submission) to include authors Monisha Sharma MPH, PhD, Tecla Temu, MD, PhD, Sarah Masyuko, MBChB, MPH, PhD, John Kinuthia, MMed, MPH, Alfred Osoti, MBChB, MMed, PhD, Jerry Zifodya, MD, MPH, Damalie Nakanjako, MMed, PhD, Anne Njoroge, MBChB, MPH, Amos Otedo, MD, PhD, Stephanie Page, MD, PhD and Carey Farquhar, MD, MPH

Additional Editor Comments (if provided):

Reviewers' comments:

Reviewer's Responses to Questions

**Comments to the Author**

1. Is the manuscript technically sound, and do the data support the conclusions?

Reviewer #1: Yes

Reviewer #2: Partly

2. Has the statistical analysis been performed appropriately and rigorously? 

Reviewer #1: Yes

Reviewer #2: Yes

3. Have the authors made all data underlying the findings in their manuscript fully available?

Reviewer #1: Yes

Reviewer #2: No

4. Is the manuscript presented in an intelligible fashion and written in standard English?

Reviewer #1: Yes

Reviewer #2: Yes

5. Review Comments to the Author

Reviewer #1: In this study, the authors measured the prevalence of hypertension in PWH and healthy individuals to understand how ART correlates with hypertension. The authors found that PWH had protection from hypertension and performed multivariate analyses between risk factors and hypertension. While these observations confirm previous studies, the indirect effect of ART on other conditions such as hypertension in PWH remains to be clarified.

Overall, the data appear to be diligently obtained and transparently described and are an important contribution to the complex area of HIV-1 therapy. Some clarifications and corrections are requested.

1. Please consider rephrasing the title: may be more appropriate and specific

2. Please consider adding the exclusion criteria in the method section

3. Line 181: Consider rectifying a typo “[IQR] 45,53” with a median age of 39.5

4. I wondered if there is a reason for not including multivariate analyses for some risk factors in Tables 3 and 4.

Reviewer #2: Authors have done a great job. However, some areas of the study need review. I have provided my view about your manuscript. Please consider re-writing your introduction in such a way that will make the readers to show great desire in reading the entire manuscript. You need to provide more information about your sampling and the preservation type used in sample collection must be mentioned. Also there are few things you need to include in your statistics section and i have made few suggestions which i think can make your work better.

6. PLOS authors have the option to publish the peer review history of their article (what does this mean?). If published, this will include your full peer review and any attached files.

Reviewer #1: **Yes: **Himanshu Batra

Reviewer #2: **Yes: **Dr. OLORUNTOBA AYODELE EKUN

---

## [Author Response · Author response to Decision Letter 0]

6 Sep 2021

Ref. Number: PONE-D-21-10329

Dear PLOS ONE Editors and Reviewers,

Thank you very much for the opportunity to revise and resubmit our manuscript, entitled “Factors associated with hypertension prevalence and correlates among adults with and without HIV in Western Kenya”. We have revised our manuscript in response to the comments made by reviewers. In this cover letter, we provide a point-by-point response to the reviewers’ comments. We are returning two copies of the revision, one in which the changes are tracked (“tracked”) and the other in which all changes have been accepted (“clean”).

We greatly appreciate your consideration and look forward to hearing from you.

Sincerely, 

Jerusha Mogaka (corresponding author)

Thank you for reviewing the manuscript. We have addressed the comments as follows:

Editorial Comments

Journal Requirements:

– Please ensure that your manuscript meets PLOS ONE's style requirements, including those for file naming. The PLOS ONE style templates can be found at https://journals.plos.org/plosone/s/file?id=wjVg/PLOSOne_formatting_sample_main_body.pdf and https://journals.plos.org/plosone/s/file?id=ba62/PLOSOne_formatting_sample_title_authors_affiliations.pdf

Response: We have reviewed the manuscript to ensure it meets PLOS One style requirements

– In your Methods section, please provide additional information about the:

a) participant recruitment method 

Response: We have now added the following to the methods section:

Line 121-122: “We utilized stratified random sampling to ensure an equal number of males and females since male sex is associated with higher CVD risk.”

Participants seeking services at the HIV voluntary testing sites after they tested HIV negative were invited to participate in the study while PWH were recruited from the Comprehensive Care Center when they presented for their routinely scheduled visits at the Kisumu County Hospital. This has been included in Line 131-135).

b) the demographic details of your participants. 

Response: We enrolled 298 HIV negative and 300 PWH, with an equal distribution of male and female participants in both groups.(Line 199-201). Demographic details have been included in Table 1. (Line 218). 

Please ensure you have provided sufficient details to replicate the analyses such as:

a) the recruitment date range (month and year), 

Response: We recruited participants between September 2017- May 2018. This has been updated in Line 116

b) a description of any inclusion/exclusion criteria that were applied to participant recruitment, 

Response: We have edited the inclusion and exclusion criteria to include the following:

Line 130 -137

 “Eligible study participants had to be �30 years old, living within a 50 km radius of the hospital, and seeking routine services at KCH. Additionally, HIV-negative persons recruited from voluntary HIV counseling and testing (VCT) services at KCH had to screen negative for a rapid HIV test while PWH were to be in care at an HIV comprehensive care clinic (CCC) and on ART for �6 months. 

We excluded from the study individuals not willing to give informed consent or participate in any of the study procedures, not willing to screen for HIV for those who verbalized they were HIV negative and PWH who were on ART for <6 months.” (Line 140-142).

c) a table of relevant demographic details

Response: Table 1 provides demographic details of all individuals included in our analysis along with behavioral factors (alcohol, smoking, fruit/vegetable intake) (Line 218).

d) a statement as to whether your sample can be considered representative of a larger population, 

Response: We have now added the following to the Study design and setting section:

Line 117-121: “We selected KCH as our recruitment site since its patient population of 12,903 PWH covers a large catchment area around 59,000 which increases the likelihood of enrolling a representative sample. Although we recruited from a hospital, we enrolled HIV-negative participants from voluntary HIV counseling and testing (VCT) services as these individuals were likely representative of the healthy general population.”

e) a description of how participants were recruited, and 

Response: Participants seeking services at the HIV voluntary testing sites after they tested HIV negative were invited to participate in the study while PWH were recruited from the Comprehensive Care Center when they presented for their routinely scheduled visits at the Kisumu County Hospital. This has been included in Line 132-135).

f) descriptions of where participants were recruited and where the research took place. 

Please note that according to our policies, if materials, methods, and protocols are well established, authors may cite articles where those protocols are described in detail, but the submission should include sufficient information to be understood independent of these references (https://journals.plos.org/plosone/s/submission-guidelines#loc-materials-and-methods).

Response: We have updated the methods section of the proposal to include the above requested details on participants’ inclusion and exclusion criteria, recruitment method and study site.

Line 130-147 which read “ Inclusion criteria for all participants were being �30 years old, living within a 50 km radius of the hospital, seeking routine services at KCH, and willing to provide informed consent. Eligibility criteria for PWH were being enrolled in care at an HIV comprehensive care clinic and on ART for �6 months. Eligibility criteria for HIV-negative persons were seeking voluntary HIV testing and counseling services at KCH and screening negative for a rapid HIV test. We excluded from the study individuals not willing to give informed consent or participate in any of the study procedures, not willing to screen for HIV for those who verbalized they were HIV negative and PWH who were on ART for <6 months.” 

– Please amend the manuscript submission data (via Edit Submission) to include authors Monisha Sharma MPH, PhD, Tecla Temu, MD, PhD, Sarah Masyuko, MBChB, MPH, PhD, John Kinuthia, MMed, MPH, Alfred Osoti, MBChB, MMed, PhD, Jerry Zifodya, MD, MPH, Damalie Nakanjako, MMed, PhD, Anne Njoroge, MBChB, MPH, Amos Otedo, MD, PhD, Stephanie Page, MD, PhD and Carey Farquhar, MD, MPH

Response: Thank you for your suggestion. We have amended the manuscript submission data via Edit Submission.

Reviewers' comments:

Reviewer #1 

General Comments: 

– Please consider rephrasing the title: may be more appropriate and specific.

Response: We have revised the title of the manuscript to “Factors associated with hypertension prevalence and correlates among adults with and without HIV in Western Kenya”

– Please consider adding the exclusion criteria in the method section 

Response: We have now defined the exclusion criteria in the study procedures section in line 135-137. “We excluded from the study individuals not willing to give informed consent or participate in any of the study procedures, not willing to screen for HIV for those who verbalized they were HIV negative, and PWH who were on ART for <6 months.”

– Line 181: Consider rectifying a typo “[IQR] 45,53” with a median age of 39.5 

Response: We thank the reviewer for the careful reading of our manuscript. We have now rectified the typo error in line 201-202. “The median age was 45 (interquartile range [IQR] 39.5, 53) years in PWH and 40 (IQR: 31, 55) for the HIV-negative participants (Table 1).”

– I wondered if there is a reason for not including multivariate analyses for some risk factors in Tables 3 and 4.

Response: For the multivariate analyses both in tables 3 and 4 we only included risk factors that were identified a priori as potential confounders of interest that have been identified in previously published literature. This is also to avoid over adjusting. In addition, some risk factors from tables 3 and 4 were not statistically significantly associated with the outcome (hypertension). Finally, sex was not included since we utilized stratified random sampling to balance the confounding due to sex.

Reviewer #2: 

– Please consider re-writing your introduction in such a way that will make the readers to show great desire in reading the entire manuscript. 

Response: We have now substantially re-written the introduction section.

– You need to provide more information about your sampling and the preservation type used in sample collection must be mentioned.

Response: We include the following description of the lab procedures and have cited a prior publication for detailed laboratory procedures:

Line 151-155

Fasting blood samples were collected in red top tubes, processed and serum was extracted stored at Kenya Medical Research Institute Centers for Disease Control and Prevention (KEMRI/CDC) lab in Kisumu, Kenya. Testing for CD4 T-cell count and viral load for PWH were collected in EDTA tubes, centrifuged to extract plasma and processed locally at the KEMRI/CDC lab. A detailed description of laboratory procedures is given elsewhere has been previously described (24).

# PLOS ONE Reviewer’s Comments

– From the introduction, it was clear that the prevalence of hypertension in Kenya has been well researched and published as shown by references 4,9,10. One therefore wondered on what the essence of this present study aim at achieving is.

Response: Although previous studies have evaluated hypertension prevalence in Kenya, the results are varied—similar to estimates of hypertension prevalence in SSA in general. Therefore, the hypertension burden in SSA is not well established (1). In addition, the prevalence of hypertension among persons living with HIV who are virally suppressed on ART is not well evaluated. We have added more information on this rationale in the introduction section.

– There is a need for the authors to consider re-writing the introduction as the current one is not catching or appealing enough.

Response: We have now substantially re-written our introduction section.

– Methods: Did authors actually use data from 300 HIV-negative adults (Line 114)? See line 62 of abstract. 

Response: We thank the reviewer for catching this. While the main study collected information on 300 HIV negative two participants were later excluded from the study due to missing anthropometric and blood test results. For this study, we therefore analyzed data for 298 HIV negative adults. Line 114 has been updated and now reads: 

“We used data from a cross-sectional study of 300 PWH and 298 HIV-negative adults”

– Line 135: was the blood sample for CD4 T lymphocyte and viral load not preserved using anticoagulant? Please throw more light on this. 

Response: EDTA anticoagulant tubes were used to collect blood samples for CD4 T lymphocyte and viral load processing. This was to ensure the samples did not clot.

We have now added the following to the manuscript: “Testing for CD4 T-cell count and viral load for PWH were collected in EDTA anticoagulant tubes, centrifuged to extract plasma and processed locally at the KEMRI/CDC lab.” Line 153-154.

– Line 160: How did the authors arrive at the cut off value of >130mg/L for LDL-C (As against the well-known cut off of >100mg/dl)? 

Response: While the well-known cut off for LDL is >100mg/dl, for individuals who have no health issues LDL levels of less than 130mg/L are considered acceptable. In our study population, we recruited relatively stable individuals who were coming to the facility for routine HIV screening at the voluntary counseling and testing center and people with HIV who were stable on ART with no previous known history of heart disease. Additionally, the Kenyan guidelines on LDL cut-offs is This therefore guided our decision of using a cut-off value of >130mg/L.

– Statistics: Line 168. How did the authors test for normality of the continuous variables? 

Response: We plotted histograms and normal distribution curves to test for normality to determine the percentage of outliers. We however were not concerned with normality of the data due to the large sample size of 598 participants. With a large sample size, the distribution of mean values is approximately normally distributed even if the data themselves are not normally distributed.

– I suggest that authors should stick to one p value (e.g. p <0.05 or p<0.001) for the purpose of discussion of the results. 

Response: Thank you for the recommendation. We presented our results in different p-values to be able to show the strength of association between variables.

– Authors should consider presenting some of the results in figures.

Response: We agree that figures add visual appeal to the manuscript. However, due to the large amount of data presented, including the multivariate regressions, we feel that tables would be most appropriate to showcase our results. 

– Critically looking at the first two lines of discussion (line 241-242), the message is the same with the first line in 2nd paragraph. (Line 249). This should be reviewed.

Response: We have amended the first paragraph of the discussion was to give a summary of the study findings. 

We have reviewed and updated the 2nd paragraph to read “The higher prevalence of hypertension among the HIV-negative individuals compared to PWH, did not support the hypothesis that PWH are at increased risk of hypertension because of increased the HIV virus and ART use leading to immune activation” (Line 273-275).

– Line 282-290 discussed what was never presented as part of the results. There was nowhere in the results that the blood pressure values of the participants were documented. How do I therefore believe the discussion?

Response: These results are presented in the methods section. In line 222-227, we report results on hypertension whereby 129 individuals were found to have hypertension. Of these 71 had a previous diagnosis of hypertension. Focusing on those with a previous diagnosis, only 23 (43%) were on anti-hypertensive medication. 

– The references: The references in this section did not follow any style. I suggest that authors should follow the recommended style by the journal.

Response: We have amended the references to the recommended Vancouver style as suggested. 

Reference:

1. Dzudie A, Hoover D, Kim H-Y, Ajeh R, Adedimeji A, Shi Q, et al. Hypertension among people living with HIV/AIDS in Cameroon: A cross-sectional analysis from Central Africa International Epidemiology Databases to Evaluate AIDS. PLoS One [Internet]. 2021 Jul 1 [cited 2021 Sep 1];16(7):e0253742. Available from: https://journals.plos.org/plosone/article?id=10.1371/journal.pone.0253742

---

## [Decision Letter · Decision Letter 1]

8 Oct 2021

PONE-D-21-10329R1Factors associated with hypertension prevalence and correlates among adults with and without HIV in Western KenyaPLOS ONE

Dear Dr.Jerusha Nyabiage Mogaka

Thank you for submitting your manuscript to PLOS ONE. After careful consideration, we feel that it has merit but does not fully meet PLOS ONE’s publication criteria as it currently stands. Therefore, we invite you to submit a revised version of the manuscript that addresses the additional points reviewer #2 raised.

We look forward to receiving your revised manuscript.

Kind regards,

Weijing He, M.D.

Academic Editor

PLOS ONE

Journal Requirements:

Reviewers' comments:

Reviewer's Responses to Questions

**Comments to the Author**

1. If the authors have adequately addressed your comments raised in a previous round of review and you feel that this manuscript is now acceptable for publication, you may indicate that here to bypass the “Comments to the Author” section, enter your conflict of interest statement in the “Confidential to Editor” section, and submit your "Accept" recommendation.

Reviewer #1: All comments have been addressed

Reviewer #2: (No Response)

2. Is the manuscript technically sound, and do the data support the conclusions?

Reviewer #1: Yes

Reviewer #2: Yes

3. Has the statistical analysis been performed appropriately and rigorously? 

Reviewer #1: Yes

Reviewer #2: Yes

4. Have the authors made all data underlying the findings in their manuscript fully available?

Reviewer #1: Yes

Reviewer #2: Yes

5. Is the manuscript presented in an intelligible fashion and written in standard English?

Reviewer #1: Yes

Reviewer #2: Yes

6. Review Comments to the Author

Reviewer #1: (No Response)

Reviewer #2: (No Response)

7. PLOS authors have the option to publish the peer review history of their article (what does this mean?). If published, this will include your full peer review and any attached files.

Reviewer #1: **Yes: **Himanshu Batra

Reviewer #2: No

---

## [Author Response · Author response to Decision Letter 1]

15 Nov 2021

Ref. Number: PONE-D-21-10329

Dear PLOS ONE Editors and Reviewers,

Thank you very much for the opportunity to revise and resubmit our manuscript, entitled “Prevalence and factors associated with hypertension among adults with and without HIV in Western Kenya.” We have revised our manuscript in response to the comments made by reviewers. In this cover letter, we provide a point-by-point response to the reviewers’ comments. We are returning two copies of the revision, one in which the changes are tracked (“tracked”) and the other in which all changes have been accepted (“clean”).

We greatly appreciate your consideration and look forward to hearing from you.

Sincerely, 

Jerusha Mogaka (corresponding author)

TITLE: Prevalence and factors associated with hypertension among adults with and without HIV in Western Kenya

Journal Requirements:

– Please review your reference list to ensure that it is complete and correct. If you have cited papers that have been retracted, please include the rationale for doing so in the manuscript text, or remove these references and replace them with relevant current references. Any changes to the reference list should be mentioned in the rebuttal letter that accompanies your revised manuscript. If you need to cite a retracted article, indicate the article’s retracted status in the References list and also include a citation and full reference for the retraction notice.

Response: Thank you for the suggestion. We have reviewed the reference list and found none of the references has been retracted. However, we have replaced line 437 (reference 27) with the current WHO guidelines on physical activity.

“Bull FC, Al-Ansari SS, Biddle S, Borodulin K, Buman MP et al. World Health Organization 2020 guidelines on physical activity and sedentary behaviour. British Journal of Sports Medicine 2020;54:1451-1462. doi.org/10.1136/bjsports-2020-102955”

Editorial Comments

Reviewer #2

– I have read the revised version of the manuscript. Authors have addressed majority of issues raised. However I am still of opinion that authors should present some of the results in figure. Also all the tables should be properly presented by having only 3 horizontal parallel lines. The current table format does not fit into scientific table presentation.

Response: As per your recommendations, we have presented some of our results as a figure (line 228). We have also formatted all tables to reflect a scientific presentation table

– Title Re-wording

After consultation with the authors, we have reworded the title to remove repetition as “factors associated” and “correlates” are similar terms. 

The title of our manuscript is “Prevalence and factors associated with hypertension among adults with and without HIV in Western Kenya.”

---

## [Editor Report · Decision Letter 2]

23 Dec 2021

Prevalence and factors associated with hypertension among adults with and without HIV in Western Kenya

PONE-D-21-10329R2

Dear Dr. Jerusha Nyabiage Mogaka,

First of all, my apologies for the long review process due to many of us dealing with COVID in particularly the recent new variant.

We’re pleased to inform you that your manuscript has been judged scientifically suitable for publication and will be formally accepted for publication once it meets all outstanding technical requirements.

Kind regards,

Weijing He, M.D.

Academic Editor

PLOS ONE
---

## [Editor Report · Acceptance letter]

30 Dec 2021

PONE-D-21-10329R2 

Prevalence and factors associated with hypertension among adults with and without HIV in Western Kenya 

Dear Dr. Mogaka:

I'm pleased to inform you that your manuscript has been deemed suitable for publication in PLOS ONE. Congratulations! Your manuscript is now with our production department. 

Kind regards, 

on behalf of

Dr. Weijing He 

Academic Editor

PLOS ONE